# What Is the Benefit of Ramped Pulse Shapes for Activating Auditory Cortex Neurons? An Electrophysiological Study in an Animal Model of Cochlear Implant

**DOI:** 10.3390/brainsci13020250

**Published:** 2023-01-31

**Authors:** Elie Partouche, Victor Adenis, Pierre Stahl, Chloé Huetz, Jean-Marc Edeline

**Affiliations:** 1Jean-Marc Edeline Paris-Saclay Institute of Neurosciences (Neuro-PSI), CNRS UMR 9197, Universite Paris-Saclay, Campus CEA Saclay, Route de la Rotonde Bâtiment 151, 91400 Saclay, France; 2Departement of Scientific and Clinical Research, Oticon Medical, 06220 Vallauris, France

**Keywords:** electrical pulses, electrophysiological recordings, auditory cortex, guinea pig

## Abstract

In all commercial cochlear implant (CI) devices, the activation of auditory nerve fibers is performed with rectangular pulses that have two phases of opposite polarity. Recently, several papers proposed that ramped pulse shapes could be an alternative shape for efficiently activating auditory nerve fibers. Here, we investigate whether ramped pulse shapes can activate auditory cortex (ACx) neurons in a more efficient way than the classical rectangular pulses. Guinea pigs were implanted with CI devices and responses of ACx neurons were tested with rectangular pulses and with four ramped pulse shapes, with a first-phase being either cathodic or anodic. The thresholds, i.e., the charge level necessary for obtaining significant cortical responses, were almost systematically lower with ramped pulses than with rectangular pulses. The maximal firing rate (FR) elicited by the ramped pulses was higher than with rectangular pulses. As the maximal FR occurred with lower charge levels, the dynamic range (between threshold and the maximal FR) was not modified. These effects were obtained with cathodic and anodic ramped pulses. By reducing the charge levels required to activate ACx neurons, the ramped pulse shapes should reduce charge consumption and should contribute to more battery-efficient CI devices in the future.

## 1. Introduction

For several decades, the cochlear implant (CI) has been a neuro-prosthetic device that allowed thousands of patients affected with severe to profound hearing loss to recover hearing sensations and speech understanding. The CI users perceive relatively well speech in quiet environments, but face important challenges when placed in noisy environments such as public transportation or restaurants. Potentially, these limitations stem from the inherent large spread of current diffusing in the perilymph of the cochlea when the electrodes are activated. The consequence is that adjacent electrodes activate largely overlapping pools of auditory nerve fibers, contributing to decrease CI user’s performances [1,2]. Several methods have been proposed for reducing the spread of activation when delivering electrical pulses in the perilymph.

First, the stimulation mode (i.e., the configurations of active and return electrodes) has been, and is still, investigated to limit the current spread within the cochlea. Different stimulations were tested all aiming at focusing the stimulation current such as the bipolar mode [3,4] or the tripolar mode [5,6,7,8,9]. Second, changing the shape of electrical pulses has been proposed as a way of reducing the current spread. In commercial CI devices, the shape of electrical pulses is usually a rectangular shape (often called “biphasic square pulse”) and consists of two phases of opposite polarity to ensure charge-balanced stimulation and avoid long-term tissue damage when using monophasic pulses [10,11]. More than a decade ago, the asymmetrical pulse was proposed as the first way to modify the pulse shape. With this shape, the polarity sensitivity has been studied [12,13,14,15] and so has its relationship with neural survival [16,17,18]. Other experiments have studied its consequences on psychoacoustic performance [19,20] or on the improvements of CI fitting [21].

More recently, non-rectangular shapes have been proposed as alternative shapes. Initial results have suggested that ramped pulses can be a more selective way to activate the auditory nerve fibers than rectangular pulses [22]. The arguments for proposing a ramped shape pulse stimulation strategy rely on: (i) on physiological characteristics for activating spiral ganglion neurons (SGNs) and (ii) on current diffusion in an aqueous media. It was proposed that ramped pulses better match the ion channel dynamics of SGN than rectangular pulses [22]. Indeed, like other neurons in the auditory system, SGN neurons express low-threshold potassium channels (e.g., see [23]). These channels are sensitive to the rate of change of synaptic input currents and regulate firing patterns, in particular in adaptation to high stimulation rate, by extending the refractory period and modulating the exact action potential timing [23,24,25,26,27]. More specifically, by using patch clamp recordings from cultured SGNs, it was shown that to trigger the emission of action potentials, the ramped pulses needed to reach a higher peak amplitude than the rectangular pulses (e.g., 7nA vs. 5nA in [22]). It was also envisioned that ramped pulses can have a more restricted diffusion pattern in the perilymph fluid than the rectangular pulses. This potentially relates to the principle that stimulation in aqueous media results in diffusion of the electrical current [28]. In the case of a square stimulus, the peak value attenuates with increasing distance, but the square shape is preserved and, hence, its infinite slope can activate a large pool of neurons. If the electrical stimulation is performed with a ramped pulse, both the slope and the peak value are attenuated with increasing distance from the point of stimulation, which can reduce the number of activated neurons (Figure 3 in ref. [22]).

Two recent studies have investigated the consequence of using ramped pulses in vivo, either on physiological measurements or on psychophysical performances. Quantifying electrically evoked brainstem responses (eABR) in mice, Navntoft and colleagues (2020) [29] reported that anodic or cathodic ramped pulses elicited eABR responses with lower thresholds and steeper growth functions than anodic or cathodic rectangular pulses. These findings, in apparent contradiction with the Ballestero et al. (2015) [22] findings, suggested that the maximal response can be obtained with less charge when ramped pulses are used instead of rectangular pulses. Among the different ramped pulses they evaluated, the ramp-UP (with increasing amplitude over time) seemed to be more efficient than ramp-DOWN (with decreasing amplitude over time) to generate steeper slopes of eABRs growth functions (see Figure 2 in ref. [29]). In human subjects, slightly lower thresholds were found for ramped pulses in a single channel detection task, but there was no significant difference between ramp-UP and ramp-DOWN across all tested situations [30].

In the present study, we further explore the physiological consequences of using ramped pulse shapes to better understand the links between the physiological results in animal and the human psychophysical results. In vivo extracellular recordings were obtained from auditory cortex neurons of anesthetized guinea pigs, and the growth functions obtained for rectangular pulses (anodic and cathodic) and four different shapes of ramped pulses (anodic and cathodic) were quantified. We choose to quantify auditory cortex responses rather than evoked responses from the auditory brainstem structures because there are potentially more direct relationships between perceptual performance and auditory cortex responses [31,32,33,34] than with responses from subcortical structures (but see [35] for opposite results in noisy conditions).

## 2. Methods

### 2.1. Subjects

The animals were 24 pigmented guinea pigs (*Cavia Porcellus*) from 3 to 19 months old and weighting between 487 to 1238 g. They had a heterogeneous genetic background and came from our own colony housed in a humidity (50–55%) and temperature (22–24 °C) controlled facility on a 12/12 h light/dark cycle (light on at 7:30 A.M.) with free access to food and water. Accredited veterinarians from the Essonne District regularly checked the animal facility. The experiments were performed under the national license A-91-557 (project 2020-20, authorization 26243) and using the procedures No. 32-2011 and 34-2012 validated by the Ethic committee of our institute (CEEA59, Paris-Centre et Sud). All surgical procedures were performed in accordance with the guidelines established by the European Communities Council Directive (2010/63/EU Council Directive Decree). The animals’ audiograms were determined 3 days before cochlear implantation by testing auditory brainstem responses (ABR) under isoflurane anesthesia (2.5%) as previously described [36,37]. Some guinea pigs (*n* = 5) had modest hearing loss (20 dB in the worse cases) corresponding to their age [38,39].

### 2.2. Cortical Surgery and Cochlear Implantation

The surgery was performed under general anesthesia induced by urethane (1.2 g/kg, i.p.) and supplemented by lower doses (0.5–0.7 g/kg) when reflex movements were observed after pinching the hind paw (this reflex movement was tested every 30 min). Each animal was initially placed in a stereotaxic frame for the craniotomy. A heating blanket allowed the maintaining of the animal’s body temperature around 37 °C. After injection of a local anesthetic (xylocaine 2%, s.c.), the skin was opened, and the temporal muscles were pushed on the side. The skull was cleaned then dried, and three stainless steel screws were threaded into burr holes in the calvarium to anchor a subminiature socket embedded in dental acrylic cement. A craniotomy was performed on the left temporal bone, 5 mm behind the Bregma on the rostro-caudal axis to expose the primary auditory cortex; the opening was 8–10 mm wide. The skin behind the right pinna was opened and the tympanic bulla was exposed. The bulla was opened under binocular control with a 2 mm cutting burr (mounted on a surgical drill) and the cochlea orientation was determined based on anatomical landmarks (round window). A cochleostomy was performed by hand around 1 mm below the round window with a 0.4 mm diameter trephine (see details in [37]). Two separate cables were connected to the subminiature socket embedded in the cement. One consisted of a 70 mm cable ending with a large surface ball Platinum-Iridium electrode used as extra-cochlear ground. The other was the intracochlear stimulating array, a shortened version of the EVO electrode array used by Oticon Medical (Smørum, Denmark, see ref. [40]). The array (300 µm in diameter) was composed of 6 ring-shaped Platinum-Iridium electrodes, each with a 0.0046 mm² surface. Center-to-center inter-electrode distance was 600 µm. The ground electrode was inserted below the skin between the animal’s shoulders and the electrode-array was placed in front of the opened tympanic bulla. It was anchored to the muscles next to the bulla with suturing. The array supporting the six intra-cochlear electrodes was then inserted into the scala tympani. A visual confirmation of the number of inserted electrodes in the cochlea was made by direct observation through the binocular microscope. In all cases, five electrodes were inside the cochlea with the sixth being on the edge of the cochleostomy. Electrode impedances were measured before starting the stimulations and always ranged from 2 to 4 kΩ to check the post-insertion electrode interface integrity. Recordings of electrically evoked compound action potentials (eCAP) were performed immediately post-surgery while stimulating the most apical electrode with the rectangular pulses only (triggering eCAP with ramped pulses was not satisfactory). Note that, as the animals were part of another experiment which involved quantifying their residual hearing [37], they were not deafened prior to the cochlear implantation and during data acquisition. This also allows comparing the cortical responses evoked by the electrical pulses and the responses evoked by pure tones (Partouche et al., in preparation).

### 2.3. Responses of Auditory Cortex Neurons

Neural activity was recorded in the left primary auditory cortex (A1). Methodologies and procedures for data acquisition were identical to those described in our previous studies [36,41,42,43]. A 16-channels electrode matrix (ø: 33 µm, <1 MΩ), composed of two rows of 8 electrodes separated by 1000 µm (350 µm between electrodes of the same row), was inserted in A1, perpendicularly to the cortical surface to record multi-unit activity at a depth of 500–600 µm (corresponding to layer III according to Wallace and Palmer, 2008 [44]). A small silver wire (ø: 200 µm), used as a ground, was inserted between the temporal bone and the dura matter on the ipsilateral side. The location of A1 was estimated based on the pattern of vasculature observed in previous studies [36,37,42,43,45]. The signal was amplified 10,000 times (TDT Medusa, TDT, Alachua, FL, USA) and processed by an RX5 multichannel data acquisition system (TDT). The signal collected from each electrode was filtered (610–10,000 Hz) to extract multi-unit activity (MUA). A trigger level was set for each electrode to select the largest action potentials from the signal. Careful on-line and off-line examination of the waveforms indicated that the MUA collected in our experiments was most likely made of action potentials generated by 2 to 6 neurons at the vicinity of the electrode. Usually, the noise level of the recording was below 50 µV and the signal-to-noise ratio was at least 2/1 (action potential amplitude of at least 100 µV). For each experiment, the position of the electrode array was set in such a way that the two rows of eight electrode sampled neurons, responded from low to high frequency, when progressing in the rostro-caudal direction (see examples of tonotopic gradients recorded with such arrays in Figure 1 of Gaucher et al., 2012 in ref. [46] and in Figure 6A of Occelli et al., 2016 in ref. [47]). To ensure an optimal placement of the electrode array in the area AI, tonal frequency response areas (FRA) were measured by presenting 50 ms pure tones at 2 Hz, from 0.25 to 36 kHz and from 5 to 75 dB SPL.

### 2.4. Stimulation Protocols

The stimulation protocol was controlled via a research platform designed by Oticon Medical (Smørum, Denmark, designed at Vallauris, France) and connected to the implant by the subminiature socket secured on the animal’s head. The research platform embeds the stimulation chip used in the Neuro Zti implant from Oticon Medical. This allowed the study to deliver pulse ramps usable on actual commercially available cochlear implant systems. The pulse waveform is double active, generating current waveforms either anodic-first or cathodic-first as needed during the protocol. Single pulses were delivered at 4 Hz via the most apical electrode to avoid the occurrence of electrophonic responses that could contaminate electrically evoked responses, which have been described earlier with 25 Hz stimulations by Stypulkowski and van den Honert (1984) [48], and more recently by Sato et al. (2016, 2017) [49,50], in the inferior colliculus. We used a monopolar mode of stimulation, which reduces the possibility to observe contamination by electrophonic responses, in contrary to the bipolar mode (see [49,50]). The protocol used to evaluate the growth functions of auditory cortex neurons always started with rectangular pulses having a cathodic first-phase presented at 20 charge levels, from 3 to 31.5 nC with 32 repetitions at each charge level. Note that the corresponding rectangular shape shall be partially ramped-like shape at the pulse onset due to the capacitive couple of the electrode-tissue impedance. Then, four types of ramped pulses (Figure 1) were randomly presented at the same 20 levels of charge, all of them with a cathodic first phase. For two of these ramped pulses, the increase in charge level was performed by keeping the slope at a fixed value (85° and 80°), which is possible by increasing the duration and the peak amplitude of the pulses. These two types of ramped pulses will be named Fixed Slope and their abbreviations will be FS85-C and FS80-C (the C stands for the cathodic first phase). For the two other ramped pulses, the increase in charge level was performed by keeping the peak amplitude at a fixed value (750 µA and 500 µA) while changing the duration and slope of the pulses. Note that the pulse duration was from 4 to 42 µs when the pulse was set at 750 µA, and was from 6 to 63 µs when the pulse amplitude was set to 500 µA. These two types of ramped pulses will be named Fixed Amplitude and their abbreviations will be FA750-C and FA500-C (the C stands for the cathodic first phase). Finally, it is important to stress that the ramped current slope was not smooth but approximated in the current step size of 25 μA. This current step size was the authors’ choice, and even if it has limitations, it remains realistically implementable in current CIs since our study used a commercialized stimulation chip. The temporal resolution of the animal stimulator platform was 3–4 μs.

The exact same protocol was also performed with anodic first pulses. We also started with rectangular pulses having an anodic first-phase presented at the 20 charge levels, from 3 to 31.5 nC (32 repetitions of each charge), then we randomly presented the four types of ramped pulses with the anodic first-phase. The order of presentation between cathodic and anodic first protocols was randomized across animals. The different pulse shapes used in our study are represented in a schematic form in Figure 1.

### 2.5. Quantification of the Growth Function of Cortical Neurons

During offline analyses, the cortical evoked discharges triggered by each pulse were quantified over a 45 ms time-window starting at the pulse onset. We discarded from analysis the responses starting with latency < 9 ms after stimulus onset as the shortest latency obtained with acoustic stimuli in the guinea pig AI was 9 ms (Figure 3 in ref. [44]). The firing rate (FR, number of action potentials per second) obtained for each cortical electrode was averaged across the 32 repetitions of each pulse shape. A cortical recording was considered to generate significant responses when the evoked activity was three standard deviations (SD) above its spontaneous FR at a given charge level. The FRs obtained from each cortical recording exhibiting significant responses were averaged across animals to obtain group data. For each recording, a Matlab script automatically extracted four parameters from the growth function:The maximal evoked firing rate (MaxFR) was the maximal number of action potentials minus spontaneous activity obtained over the 32 repetitions of the 20 levels of charges. Spontaneous activity was computed over the last 100 ms of the inter-pulse interval (which is 250 ms with a 4 Hz stimulation).The charge level at which the MaxFR was obtained was automatically detected.The threshold was the lowest charge level producing a significant response above spontaneous activity plus three SD (see above). In addition, to guarantee that the threshold was not detected by chance, we requested that at least two consecutive charge levels above the threshold also generated significant evoked responses.The Dynamic range was the difference between the charge level producing the MaxFR and the threshold. We also computed another dynamic range (named DynRange80), which was the difference in charge level between the threshold and the first charge level eliciting a firing rate at 80% of the MaxFR.

### 2.6. Statistical Analyses

As in most cases, the raw data did not follow normal distributions (based upon Shapiro–Wilk tests), non-parametric tests were performed to compare the data obtained with rectangular pulses and those obtained with each type of ramped pulses. For all quantified parameters, two-tailed Wilcoxon signed-rank (WSR) tests were used with the level of *p* < 0.05 (alpha = 5%) as the threshold for the significance value. All tests were performed with Graphpad Prism (version 9).

## 3. Results

Data were obtained from 441 recordings collected in the primary auditory cortex of 24 guinea pigs. For each animal (*n* = 24), there were between one and five positions (mean: 2 positions) of the 16-channel electrode array. For each of these positions we used between 2–15 recordings in the data analyses (average 9 recordings/position). On average, about 18 recordings per animal were obtained (range: 5 to 41). In the following presentation of the results, we will first focus on the comparison between the responses evoked by one configuration of cathodic ramped pulses (FS80-C, see Figure 1) versus the cathodic rectangular pulses (Rec-C). Then, we will generalize these results to the other types of ramped pulses used in our study.

### 3.1. Ramped Pulses Induce Higher Evoked Firing Rate and Lower Threshold Than Rectangular Pulses

The raster plots presented in Figure 2 illustrate the trial-by-trial occurrence of action potentials (AP) during the 32 repetitions of the 20 charge levels for two simultaneous recordings in the primary auditory cortex. The left panel (Figure 2A) presents the responses triggered by the rectangular pulses (Rec-C) and the right panel (Figure 2B) the responses triggered by the ramped pulse (FS80-C). In both cases, the alternating colors (blue/red) represent the changes in charge level and the time window displays the responses from 5 to 45 ms after stimulus onset. For these two cortical recordings, the evoked responses were increased for most of the charge levels with the ramped pulses in comparison to the responses obtained with the Rec-C pulses. In addition, ramped pulses triggered responses at lower thresholds (represented by dashed lines) compared to rectangular pulses. Figure 2C,D displays the quantifications of the evoked firing rate as a function of injected charges for these two recordings, with the blue curves corresponding to the growth functions using the FS80-C pulses and the black curves representing the growth functions using the Rec-C pulses. It clearly illustrates that the firing rate increased with the injected charge with both pulse shapes, but these increases were much larger with ramped pulses compared to rectangular pulses. As a consequence, the maximal evoked firing rates (indicated by the arrows) were higher with the FS80-C than with the Rec-C shape. In addition, the thresholds, defined as three SD above spontaneous activity (indicated by the vertical dashed lines and the red dots on the curves), were lowered when ramped pulses were used.

### 3.2. The Four Cathodic Ramped Pulses Show Similar Differences Compared to Rectangular Pulses

The scattergrams presented in Figure 3 compare, for all recordings obtained with both shapes (*n* = 106), the four parameters derived from the responses to the ramped pulses FS80-C (y-axis) with those derived from the response to Rec-C pulses (x-axis). Figure 3A presents the maximal evoked firing rate (MaxFR) obtained with both pulse shapes. In a large number of cases (90/106), the MaxFR values obtained with FS80-C were higher than the values with Rec-C (dots above the diagonal line). The mean values displayed in the inset (Figure 3A) shows that the MaxFR was increased from 51.4 ± 4.1 AP/sec with Rec-C pulses to 62.9 ± 4.5 AP/sec with FS80-C (Wilcoxon test, *p* < 0.0001). Figure 3B represents the threshold values obtained with the two pulse shapes: A majority of dots were below the diagonal line indicating lower threshold values with ramped pulse compared to rectangular pulse. The inset shows that the mean values were statistically lower with ramped pulses (12.8 ± 0.6 nC for Rec-C vs. 10.5 ± 0.5 nC for FS80-C; Wilcoxon test, *p*-value < 0.0001). The charge value eliciting the MaxFR is presented in Figure 3C. As for the threshold, most of the dots were below the diagonal line indicating lower values for ramped pulses in comparison to rectangular pulses (22.53 ± 0.6 nC for the Rec-C vs. 20.63 ± 0.6 nC for the FS80-C, Wilcoxon test, *p*-value < 0.0061). Finally, Figure 3D shows the dynamic range values (see Methods for details). For this parameter, the dots were equally present on both sides of the diagonal line, indicating that there was no general trend for observing either larger or smaller dynamic range with ramped pulses. The mean values displayed in the inset confirmed that there was no difference between the two pulses shapes in terms of dynamic range (9.9 ± 0.7 nC for the FS80-C versus 9.5 ± 0.6 nC for the Rec-C; Wilcoxon test, *p* = 0.4034).

Figure 4 presents the mean values of parameters derived from evoked responses with the four cathodic ramped pulses compared with the rectangular pulse (the mean values, sem values and *p*-values are provided in Table 1). Figure 4A shows that in all but one case, the ramped pulses elicited higher firing rates than rectangular pulses (*p* < 0.001 for the FS80-C, the FA750-C and FA500-C; but *p* = 0.0775, ns for the FS85-C). Importantly, the mean threshold values (in nC) were systematically lower with the ramped pulses compared to the rectangular pulses (Figure 4B). In addition, the charge level (in nC) eliciting the MaxFR was lower compared to rectangular pulses (Figure 4C) except for the FS85-C pulses (see details in Table 1). As a consequence of these lower thresholds and lower levels eliciting the MaxFR, the dynamic range (Figure 4D) was either unchanged, compared with the rectangular pulses (for the FS80-C, FA750-C and FA500-C), or slightly reduced for the FS85-C (see details in Table 1). Note that the dynamic range computed with the first point eliciting 80% of the MaxFR was also unchanged when comparing the values obtained with the rectangular pulses and those obtained with the different shapes of ramped pulses (data not shown, *p* > 0.05 in all cases).

Due to the position of the stimulating electrode—at the end of the first and half turn of the guinea pig cochlea—there were more neuronal recordings responding to the electrical stimulations with CF between 8–16 kHz. However, the increase in cortical responses with the ramped pulses were detected in all frequency ranges, from the low frequency (CF below 4.5 kHz) to the medium (4.5 < CF < 12.7 kHz) and to the high frequencies (>12.7 kHz). We have checked that the facilitatory effects of the ramped pulses compared to the rectangular pulses were observed in all the 16 animals involved in these comparisons.

### 3.3. The Four Anodic Ramped Pulses Also Show Similar Differences Compared to Rectangular Pulses

Strikingly, similar results were obtained with the anodic pulses (Figure 5). First, compared with the rectangular pulses, the MaxFR were increased (Figure 5A, but this was significant only with the FS85-A and FS80-A pulses). Second, the thresholds were systematically lower with the ramped pulses (Figure 5B) and so was the charge level providing the MaxFR (Figure 5C). Lastly, the dynamic range was also either unchanged or significantly decreased (Figure 5D). These effects were observed on all the animals.

## 4. Discussion

The present study is the first to evaluate how different ramped pulse shapes modulate the responses of auditory cortex neurons. We show here that compared with classical rectangular biphasic pulses, ramped pulses generate higher evoked firing rates and lower thresholds for auditory cortex neurons. The dynamic range was unchanged since the maximal evoked firing rate (MaxFR) was reached earlier in the growth function of cortical neurons.

### 4.1. Methodological Considerations

Obviously, auditory cortex neurons are quite far away from the cochlea where the pulse shapes were manipulated, and therefore one can consider that cortical activity is an indirect reflection of the impact on auditory nerve fibers. Despite this, the manipulations of pulse shapes had a reliable impact on cortical evoked responses: Although they were not systematically significant, the results observed with the four ramped pulses were always in the same direction, both with a cathodic and an anodic first phase. As cortical activity is potentially more tightly related to perceptual performance [51,52,53,54,55] than auditory nerve activity, our results suggest that the use of ramped pulses can be of importance for improving the CI patient performance (e.g., see [30,56]).

During the last decade, several types of pulse shapes have been investigated to improve the perception of CI patients. The asymmetric pulses have been extensively tested in human psychoacoustic studies, especially to reveal polarity effects and to understand which phase of the pulses is efficient for activating auditory nerve fibers in humans [12,15,57,58,59,60]. In fact, pseudo-monophasic pulses combined with phase duration loudness coding reduced the lowest response thresholds, the spread of excitation, and also reduced the between electrode interference by 4–5% compared to biphasic pulses, which can be of importance for understanding the interaction between the two phases of biphasic pulses [61].

In their study using eABR as physiological index, Navntotf and colleagues (2020) [29] have tested other types of ramped pulses. Especially, these authors made comparisons between “ramped-UP” and “ramped-DOWN” pulses. In future studies, it will be useful to test these other types of ramped pulses on cortical responses since it is possible that a particular combination of ramped pulses and of asymmetry can promote lower thresholds together with more focus stimulation (a reduced spread of activation) and potentially an increase dynamic range.

Last, the results presented here are from normal hearing animals (or animals with modest hearing loss < 20 dB). It was claimed that in these conditions, the neural responses obtained from central structures can be the combination of both electroneural responses (resulting from direct activation of SGNs either at the somatic dendritic level, or axonal level) and electrophonic responses (resulting from activation of hair cells). However, it is important to point out that the contribution of electrophonic responses is relatively limited with monopolar stimulations, especially for neurons with high frequency CF (see Figures 4 and 5 in ref. [49]) and especially when low rates of stimulation have been used [62]. In addition, it is worth mentioning that the acute deafening that is performed in many studies does not provide the same response (especially in terms of response latencies) as the long-term deafening occasionally performed in some experiments (see Figure 11 in ref. [62]).

### 4.2. Comparison with Previous Studies Using Ramped Pulses

In their initial study, Ballestero and colleagues (2015) [22] proposed that two factors contribute to improve the spatial selectivity delivered by ramped pulses. First, it was assumed that the spatial spread of electrical currents is smaller with ramped pulses than with rectangular pulses (Figure 3 in ref. [22]), the idea being that the currents generated by ramped pulses are more attenuated in a conductive liquid (the perilymph) than the currents generated by rectangular pulses. Although this might be true in theory, this has not been clearly demonstrated yet, except in models [28,63]. Second, the biophysical properties of SGNs neurons generate a sensitivity for the rate at which depolarizing inputs reach these neurons. In fact, a sensitivity to the rate at which the input changes (i.e., the slope) is a general property observed in many neurons and is related to the dynamics of the ion channels underlying a neuron’s spiking behavior. This property has been reported in neurons from different structures such as cortical [64] or thalamic neurons [65]. In the auditory system, this has been described for octopus and bushy cells in the cochlear nucleus [24,66], for the principal neurons of the medial superior olive [67] and also from the dendrites of SGNs [68]. In the cochlear nucleus, this sensitivity relies on the activation of low threshold activated potassium channels sensitive to dendrotoxin (i.e., containing Kv1.1/2/6 subunits). It was reported that Kv1.1 and Kv1.2 subunits are strongly expressed in SGNs, where they are determinant in regulating firing patterns [23]. This strongly suggests that SGNs should be sensitive to the input slope and that, potentially, this feature is at play during electrical activation when using different stimulus shapes.

Based on these two factors, a reduction in the spread of excitation should be expected and as a consequence a global reduction of the firing rate in the auditory nerve. The effects reported by Ballestero and colleagues (2015) [22] were in this direction: The smoother was the slope of the ramped pulses, the lower was the firing probability of SGNs and the larger was the charge level required to evoke an action potential (Figure 6 in ref. [22]). Actually, reducing the slope and the peak amplitude seems to be the two key factors to reduce the firing probability of SGNs (Figure 5 in ref. [22]).

Surprisingly, our results and those described by Navntoft and colleagues (2020) [29] did not confirm these in vitro findings. Indeed, both with eABRs (Navntoft et al., 2020 [29]) and with cortical recordings (present data), the threshold for activating central auditory neurons was lower with ramped pulses than with rectangular pulses. This indicates that lower charge values are required to activate SGNs with ramped pulses than with rectangular pulses, suggesting that ramped pulses seem, in fact, more efficient to activate SGNs than rectangular pulses. In addition, our data also pointed out that the maximal firing rate was often larger with ramped pulses (see Figure 5) than with rectangular pulses, which also indicates a better efficacy of ramped pulses. In theory, the decrease in threshold values and the increase in the maximal firing rates should promote a larger dynamic range. However, since the charge level required for eliciting the maximal evoked firing was lower with the ramped pulse, the dynamic range was unchanged for most of the ramped pulse shapes (it was even slightly reduced in some cases, when the charge level eliciting the maximal evoked firing rate was more decreased than the threshold, and this effect was also detected with the DynRange80, see methods for definition). Many factors can explain the differences between the Ballestero’s results [22] and those described here and by Navntoft et al. (2020) [29]. (i) In their patch recordings, Ballestero et al. (2015) [22] evaluated the firing probability in cultured SGNs by applying a series of depolarizing current steps directly via the recording electrode. In contrast, our results and those by Nanvtoft et al. (2020) [29] were obtained by activating a pool of auditory nerve fibers and by recording either the responses of a small group of cortical neurons (here) or a global response in the brainstem (eABR), which in both cases probably reflect the synchronous activity of the most sensitive fibers to low-level electrical stimuli. Thus, many crucial steps have being bypassed in the Ballestero study [22], making it difficult to directly relate the in vitro and the in vivo results. Those steps include the electrode impedance, the fluid, bone, the electrode position relative to the peripheral processes, the site of excitation, the extent of myelination, spread of excitation, temporal, and spatial integration. (ii) In contrast to the two in vivo studies, Ballestero et al. (2015) [22] did not control for the amount of injected charge, nor did they test for declining ramped pulses to enable a fair comparison with rectangular pulses. Thus, the higher firing probability they observed with steeper slopes could be the result of more injected charge. (iii) Ballestro et al. (2015) [22] used a ramped pulse with a foot-amplitude, meaning that the beginning of the pulse still had a rectangular shape (Figure 2a in ref. [22]). Their hypothesis was that the foot-amplitude of a given amplitude starts to activate SGNs at the subthreshold level, then the ramp steepness is used to control the evoked firing rate. In contrast, both Navntoft and colleagues (2020) [29] and our study used the purely ramped stimulus in which the stimulus phases ramped up from 0 to X μA (or ramped down from X μA to 0 μA in the pulses descending ramped pulses of the Navntoft’s study [29]). To summarize, there are so many differences between the methodology used by the in vitro study [22] and by the two in vivo studies (ref. [29] and the present one) that it was almost unrealistic to expect similar results.

## 5. Conclusions: Is There the Benefit of Using Ramped Pulses in CI Prosthetics?

The present results and others [29,30] suggest that one of the key advantages of ramped pulses is its ability to decrease the threshold for activating central auditory structures. This decrease was rather modest from eABR recording (about 1 nC from Figure 2b in ref. [29]) but more substantial in our cortical data (from 2 to 6 nC depending on the type of ramped shape). Obviously, if lower charges are required for activating auditory nerve fibers, this can be crucial for the power consumption of CI devices. In fact, implementation of a non-rectangular biphasic stimulation waveform (with decaying exponential cathodic phases or growing exponential cathodic phases) can result in up to 25% charge savings and energy savings, which is crucial, in the future, for designing fully implantable CI [56].

## Figures and Tables

**Figure 1 brainsci-13-00250-f001:**
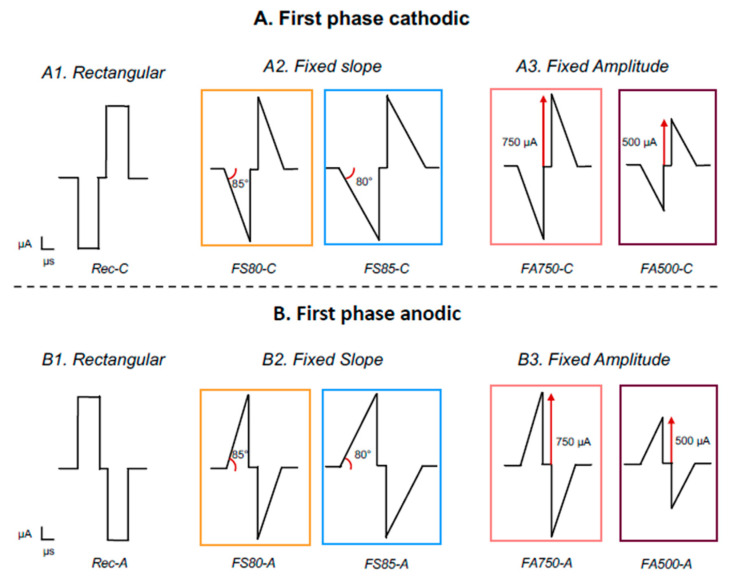
Schematic diagram representing the different pulse shapes used in this study. (**A**) Rectangular and four ramped pulses with either a fixed slope (of 85° and 80°) or a fixed peak amplitude (of 750 and 500 µA) with a cathodic-first phase. (**B**) Same pulse shapes with the anodic-first phase. The color code surrounding each ramped pulse shape will be used for the subsequent figures.

**Figure 2 brainsci-13-00250-f002:**
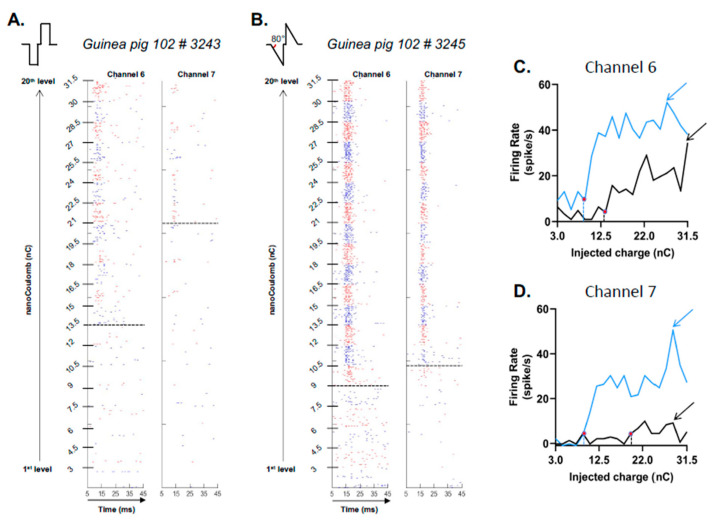
Individual examples of evoked firing rate and threshold of auditory cortex neurons with the rectangular cathodic pulses (**A**) and with one of the ramped pulse shape cathodic pulses (**B**). In (**A**,**B**), the rasters represent the raw evoked responses obtained in the auditory cortex for two simultaneous recordings (labeled channels 6 and 7) with increasing levels of charges from 3 to 31.5 nC (see the scale on the left side of each panel). In these rasters, each dot represents the occurrence of an action potential and each change from blue to red in the raster indicates an increase in charge level. The dashed lines indicate the threshold value obtained for each recording, i.e., the charge level triggering an evoked firing rate one standard deviation above spontaneous activity (see Methods for details). (**A**) Evoked responses obtained with rectangular cathodic pulses. (**B**) Evoked responses obtained with ramped cathodic pulses. Note the large increase in the number of action potentials triggered at each stimulation pulse when ramped pulses were used compared to rectangular pulses. (**C**,**D**) Quantification of evoked responses for the two channels displayed in (**A**,**B**). In both cases, the growth functions display the evoked firing rate (minus the spontaneous activity) as a function of the charge level obtained with rectangular pulses (black curve) and with the ramped pulses (blue curve). Both in (**C**,**D**) the red dots indicate the charge level considered as threshold and the arrows charge level where the maximal evoked firing rates was detected.

**Figure 3 brainsci-13-00250-f003:**
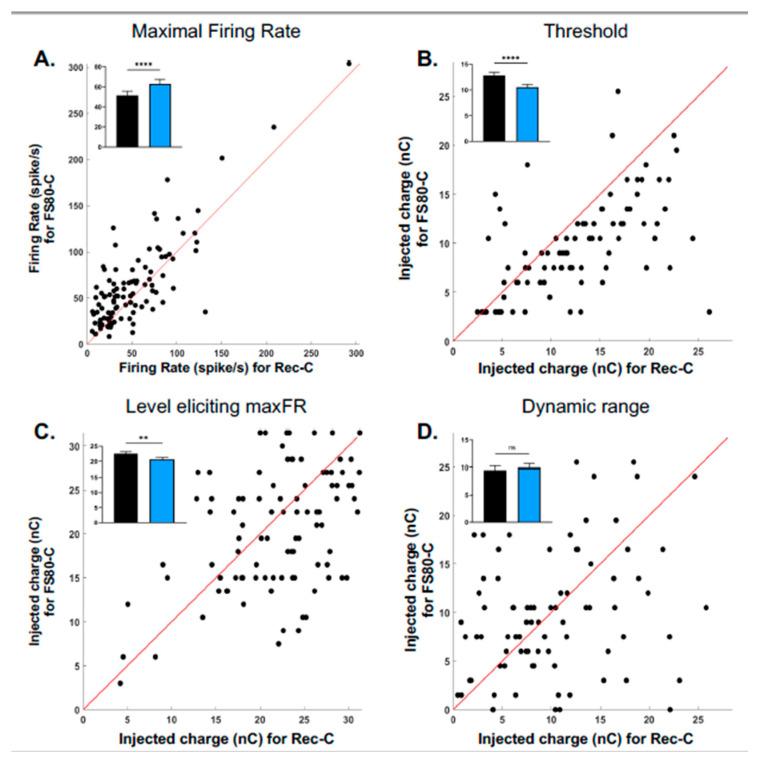
Scattergrams comparing the values obtained with ramped pulses (Fixed Slope of 80°) with those obtained with rectangular pulses on four parameters of evoked responses. For all scattergrams, each dot represents a single recording, and the red diagonal line is the equality line. (**A**) Maximal evoked firing rate (MaxFR) for the rectangular pulses vs. MaxFR for the ramped pulses. For a large number of recordings (86/106 dots above the diagonal line) the Max FR elicited by ramped pulses was higher than by the rectangular pulses. (**B**) Threshold for the rectangular pulses vs. threshold for the ramped pulses. For a large number of recordings (96/106 dots below the diagonal line) the threshold was lower with ramped pulses than with rectangular pulses. (**C**) Level eliciting the MaxFR for the rectangular pulses vs. for the ramped pulses. For a large number of recordings (76/106 dots below the diagonal line), this charge level was lower with ramped pulses than with rectangular pulses. (**D**) Dynamic Range for the rectangular pulses vs. for the ramped pulses. There was a similar number of recordings with a higher and a lower dynamic range with the ramped pulses. In each scattergram, the inset provides the mean values ± sem (** *p* < 0.01; **** *p* < 0.001; ns: not significant). Data are from 106 recordings obtained in 16 animals.

**Figure 4 brainsci-13-00250-f004:**
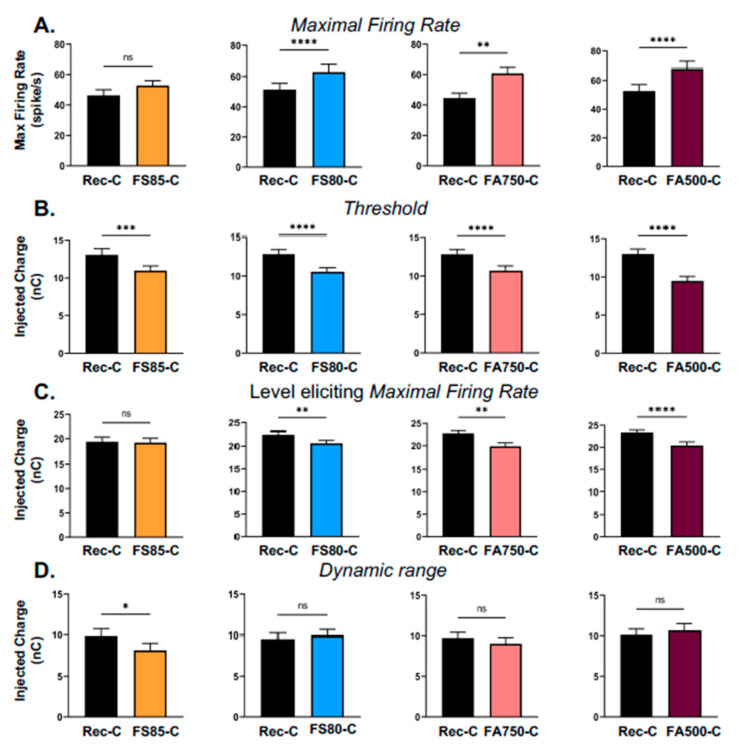
Mean values (±sem) obtained for four parameters of evoked responses with the rectangular pulses and the four types of cathodic-first ramped pulses. For each parameter, statistical comparisons (Wilcoxon tests) were made between the set of recordings tested with the rectangular pulses and with a given shape of ramped pulses. (**A**) The maximal evoked firing rate (MaxFR) was increased when stimulations were performed with the four types of ramped pulses compared to rectangular pulses. This increase was significant for three types of ramped pulses (*p* < 0.01 with FS80-C, FA750-C and FA500-C), but did not reach significance with the FS85-C ramped pulses (*p* = 0.07). (**B**) The threshold was significantly decreased when stimulations were performed with the four types of ramped pulses compared to rectangular pulses (*p* < 0.01 in all cases). (**C**) The charge level at which the MaxFR was observed was decreased when stimulations were performed with the four types of ramped pulses compared to rectangular pulses. This decrease was significant for three types of ramped pulses (*p* < 0.01 with FS80-C, FA750-C and FA500-C), but did not reach significance with the FS85-C ramped pulses (*p* = 0.8278). (**D**) The dynamic range was not significantly modified when stimulations were performed with three types of ramped pulses compared to rectangular pulses (*p* = 0.40, *p* = 0.31 and *p* = 0.92 for the FS80-C, FA750-C and FA500-C, respectively). It was significantly decreased with the FS85-C (*p* < 0.05). See Table 1 for the mean values (±sem) and the *p* values. The number of recordings and the numbers of animals involved in each comparison are provided in Table 1. (* *p* < 0.05; ** *p* < 0.01; *** *p* < 0.005; **** *p* < 0.001; ns: not significant).

**Figure 5 brainsci-13-00250-f005:**
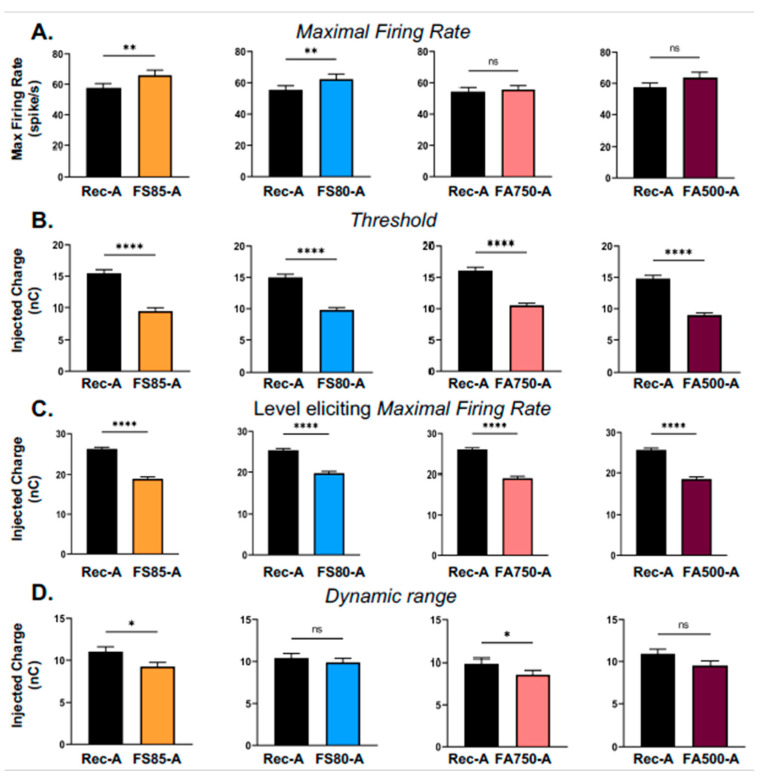
Mean values (±sem) obtained for four parameters of the evoked responses with the rectangular pulses and the four types of anodic-first ramped pulses. The conventions are the same as for Figure 4. (**A**) The maximal evoked firing rate (MaxFR); (**B**) The threshold; (**C**) The charge level at which the MaxFR was observed; (**D**) The dynamic range. Note that in general, the effects obtained with the anodic-first pulses were similar to those described with the cathodic-first pulses on Figure 4: the MaxFR was significantly increased in two out of four cases; the thresholds were significantly decreased and so were the charge levels eliciting the MaxFR. The dynamic ranges were either unchanged or slightly decreased. See Table 2 for the mean values (±sem) and the *p* values. The numbers of recordings and animals involved in each comparison are provided in Table 2. (* *p* < 0.05; ** *p* < 0.01; **** *p* < 0.001; ns: not significant).

**Table 1 brainsci-13-00250-t001:** Mean ± sem of the parameters extracted from the responses of ACx neurons with cathodic-first pulses. For each comparison, N refers to the number of animals and n refers to the number of recordings providing significant responses to both pulse shapes.

	Rec-C vs. FS85-C (N = 16 ; n = 60)	Rec-C vs. FS80-C (N = 16 ; n = 106)	Rec-C vs. FA750-C (N = 16 ; n = 85)	Rec-C vs. FA500-C (N = 16 ; n = 94)
Maximal Firing Rate	46.38 (±3.7) vs. 52.80 (±3.2)	51.43 (±4.2) vs. 62.96 (±4.6)	44.59 (±3.2) vs. 60.77 (±4.1)	52.66 (±4.5) vs. 68.13 (±4.7)
*p* value	0.0775	< 0.0001	0.0011	< 0.0001
Threshold	13.07 (±0.8) vs. 10.97 (±0.6)	12.83 (±0.6) vs. 10.56 (±0.5)	12.83 (±0.6) vs. 10.69 (±0.6)	13.01 (±0.6) vs. 9.48 (±0.6)
*p* value	0.0006	< 0.0001	< 0.0001	< 0.0001
Charge level at MaxFR	19.38 (±0.9) vs. 19.18 (±0.9)	22.53 (±0.6) vs. 20.63 (±0.7)	22.76 (±0.6) vs. 19.87 (±0.8)	23.36 (±0.6) vs. 20.41 (±0.8)
*p* value	0.8278	0.0061	0.0017	<0.0001
Dynamic range	9.83 (±0.9) vs. 8.056 (±0.9)	9.56 (±0.7) vs. 9.98 (±0.7)	9.72 (±0.7) vs. 9.02 (±0.8)	10.15 (±0.7) vs. 10.70 (±0.8)
*p* value	0.0258	0.4034	0.3157	0.9239

**Table 2 brainsci-13-00250-t002:** Mean ± sem of the parameters extracted from the responses of ACx neurons with anodic-first pulses. For each comparison, N refers to the number of animals and n refers to the number of recordings providing significant responses to both pulse shapes.

	Rec-A vs. FS85-A (N = 24 ; n = 195)	Rec-A vs. FS80-A (N = 24 ; n = 212)	Rec-A vs. FA750-A (N = 24 ; n = 186)	Rec-A vs. FA500-A (N = 24; n = 199)
Maximal Firing Rate	57.65 (±2.8) vs. 66.06 (±3.3)	55.49 (±2.6) vs. 62.35 (±3.2)	54.37 (±2.7) vs. 55.68 (±2.5)	57.51 (±2.7) vs. 63.67 (±3.5)
*p* value	0.0075	0.0045	0.6149	0.0715
Threshold	15.46 (±0.6) vs. 9.61 (±0.3)	14.99 (±0.5) vs. 9.81 (±0.4)	16.07 (±0.5) vs. 10.52 (±0.4)	14.80 (±0.5) vs. 8.98 (±0.3)
*p* value	<0.0001	<0.0001	<0.0001	<0.0001
Charge level at MaxFR	26.48 (±0.4) vs. 18.95 (±0.5)	25.35 (±0.4) vs. 19.73 (±0.5)	26.02 (±0.4) vs. 19.15 (±0.5)	25.67 (±0.4) vs. 18.47 (±0.5)
*p* value	<0.0001	<0.0001	<0.0001	<0.0001
Dynamic range	11.03 (±0.6) vs. 9.250 (±0.5)	10.41 (±0.5) vs. 9.89 (±0.5)	9.90 (±0.6) vs. 8.58 (±0.5)	10.91 (±0.6) vs. 9.5 (±0.5)
*p* value	0.0159	0.2471	0.0280	0.0601

## Data Availability

Data presented in this study are available upon request from the corresponding author.

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
