# Peer review of "What Is the Benefit of Ramped Pulse Shapes for Activating Auditory Cortex Neurons? An Electrophysiological Study in an Animal Model of Cochlear Implant"

_brainsci, 2023, doi:10.3390/brainsci13020250_

Round 1

Reviewer 1 Report

Comments and Suggestions for Authors

The paper describes an experimental evaluation of variations in stimulus waveforms for electrical stimulation of the auditory nerve. Cortical responses in guinea pigs were analyzed for four different stimuli with ramping shapes applied to implanted electrodes in the cochlea in comparison with standard rectangular biphasic pulse shapes. The ramped stimuli yielded lower charge thresholds and higher cortical firing rates than the rectangular pulses and might be advantageous for battery power reduction.

I have only minor comments regarding the description of the methods which might be addressed in the text.

L45: shape of electrical pulses -> throughout the manuscript I did not find a description of the properties of the electrical pulses. I assume that the stimulus generators produce constant current output signals and that the shape of the electrical pulses is a current waveform. It may be mentioned that the corresponding voltage for rectangular pulses will be ramped due to the capacitive component of the electrode-tissue impedance. When introducing the ramped stimulus waveforms it would be interesting to know how these ramps were generated (sampling rate, discrete steps of increasing amplitudes). Also, the pulse width of the rectangular stimulus pulses should be specified or the range of pulse width for constant amplitude pulses (assuming an amplitude of 750 uA and a range of charge per phase from 3 to 31.5 nC the pulse width would vary from about 4 to 42 us).

L80: apparent contraction -> apparent contradiction

L94-99: These two sentences are part of the results and discussion sections and should be removed from the introduction.

L246: types of ramped

Author Response

The paper describes an experimental evaluation of variations in stimulus waveforms for electrical stimulation of the auditory nerve. Cortical responses in guinea pigs were analyzed for four different stimuli with ramping shapes applied to implanted electrodes in the cochlea in comparison with standard rectangular biphasic pulse shapes. The ramped stimuli yielded lower charge thresholds and higher cortical firing rates than the rectangular pulses and might be advantageous for battery power reduction.

I have only minor comments regarding the description of the methods which might be addressed in the text.

L45: shape of electrical pulses -> throughout the manuscript I did not find a description of the properties of the electrical pulses. I assume that the stimulus generators produce constant current output signals and that the shape of the electrical pulses is a current waveform. It may be mentioned that the corresponding voltage for rectangular pulses will be ramped due to the capacitive component of the electrode-tissue impedance.

We now provide a more detailed description of the hardware, including the stimulation chip (lines 208-211).

We have also mentioned that the capacitive component of the electrode-tissue impedance partially distorted the rectangular shapes (lines 221-223).

When introducing the ramped stimulus waveforms it would be interesting to know how these ramps were generated (sampling rate, discrete steps of increasing amplitudes).

We agree that these details were missing: The ramped current slope was not smooth but approximated in current step size of 25 μA. Note that the current step size was the authors’ choice. Even if it has limitations, it remains realistically implementable in current CIs since our study used a commercialized stimulation chip. The temporal resolution of our animal stimulator platform was 3–4 μs (lines 235-239).

Also, the pulse width of the rectangular stimulus pulses should be specified or the range of pulse width for constant amplitude pulses (assuming an amplitude of 750 uA and a range of charge per phase from 3 to 31.5 nC the pulse width would vary from about 4 to 42 us).

The pulse width of the ramped pulses varied with the injected charges from 4-42µs for a peak pulse amplitude of 750µA and from 6-63µs for a peak pulse amplitude of 500µA. This is now mentioned on lines 231-233.

L80: apparent contraction -> apparent contradiction. This was corrected. Thank you !

L94-99: These two sentences are part of the results and discussion sections and should be removed from the introduction.

These two sentences were deleted.

L246: types of ramped.

This was corrected. Thank you !

Reviewer 2 Report

Comments and Suggestions for Authors

The authors studied the effect of ramped pulse shapes in relation to cochlear implant stimulation of the auditory cortex in guinea pigs. The study addresses the important topic of optimizing cochlear implant stimulation in terms of stimulation parameters. The authors found that similar cortical activation can be achieved with ramped pulse shapes to rectangular shapes. This implies that this approach may be used to reduce charge consumption in cochlear implants leading to more energy-efficient CI devices. The article is well-written and clearly structured. The introduction provides a good overall background and the figures are informative. Due to the overall good quality of the article I only have some comments, which I will address in the following:

The first issue is that is not fully clear why the authors choose to measure in the auditory cortex instead of using more robust measures assessing the lower auditory pathway. The rationale for this should be shortly addressed in more detail in the introduction.
When recording from the cortex more possibilities exist to introduce variance and affect the measurement parameters. What role did the electrode positioning play in introducing variability? What were the CFs of the measured cortical neutrons? And how was the anesthesia level controlled between conditions? Please discuss.  
Why were the guinea pigs not deafened prior to CI implantation? How does this affect the results? Explain and discuss.
Please provide more detailed information on how the 441 recordings were derived from 24 guinea pigs and 16-channel electrode arrays. Were measures repeated at different cortical positions in some animals? Please provide a more detailed overview in the method section.
Were electrode sites excluded from the study? How many?
Another issue is, that it is not clear, why all recordings were pooled across animals, and the analysis was not instead performed on the group level of 24 guinea pigs. This is not fully clear. Would the outcome have been the same? Please comment and include an example to support the claim. Or at least it should be pointed out more clearly at each instance, how many of the provided neuron numbers (n=x), were stemming from the same/different animal(s) and how many neurons stemmed from the same 16-channel array. Please comment, discuss, and include that information in more detail throughout the results.
What exactly was the trigger level for MUA extraction? Was it an absolute threshold? Please explain.

Author Response

The authors studied the effect of ramped pulse shapes in relation to cochlear implant stimulation of the auditory cortex in guinea pigs. The study addresses the important topic of optimizing cochlear implant stimulation in terms of stimulation parameters. The authors found that similar cortical activation can be achieved with ramped pulse shapes to rectangular shapes. This implies that this approach may be used to reduce charge consumption in cochlear implants leading to more energy-efficient CI devices. The article is well-written and clearly structured. The introduction provides a good overall background and the figures are informative. Due to the overall good quality of the article I only have some comments, which I will address in the following:

The first issue is that is not fully clear why the authors choose to measure in the auditory cortex instead of using more robust measures assessing the lower auditory pathway. The rationale for this should be shortly addressed in more detail in the introduction.
We choose to quantify auditory cortex responses because it is often assumed that the cortical activity is more related with perceptual performance both in humans and in animals (e.g. see Edeline & Weinberger 1993; Bieszczak & Weinberger 2010; Froemke et al. 2012; Polley et al 2006). This is now explained in the last sentence of the introduction (line 94-98)
Thus, we hope that the effects described here at the cortical level will influence the perceptual performance of CI subjects. 

When recording from the cortex more possibilities exist to introduce variance and affect the measurement parameters. What role did the electrode positioning play in introducing variability? What were the CFs of the measured cortical neutrons? And how was the anesthesia level controlled between conditions? Please discuss.  
1. We agree that the position of the electrode array on the cortical map could potentially introduce some variability in the results of the electrical stimulations. Due to the position of the stimulating electrode - at the end of the first and half turn of the guinea pig cochlea – we had more neuronal recordings responding to the electrical stimulations with CF between 8-16kHz. However, we observed that the increase in cortical responses with the ramped pulses were detected in all frequency ranges, from the low frequency (CF below 4.5kHz) to the medium (4.5<CF<12.7kHz) and to the high frequencies (>12.7kHz).  This is now indicated in the revised version on lines 411-418.

2. The level of anesthesia was checked every 30 minutes by testing the reflex to hind paw stimulation (lines 136-137). 
Note that we have recently performed long lasting experiments under urethane anesthesia at 5 levels of then auditory system (Souffi et al 2023) and the variability of evoked responses was not larger at the cortical level than in the 3 investigated subcortical levels (auditory thalamus, inferior colliculsus and cochlear nucleus). 

Why were the guinea pigs not deafened prior to CI implantation? How does this affect the results? Explain and discuss. 
The animals were not deafened prior starting the CI implantation because we aimed at comparing the strength and latencies of responses elicited in auditory cortex by the different shapes of electrical pulses with those elicited by pure tones. This will be the topic of another article as mentioned lines 169-170. 

As the animals were not deaf, it is possible that electrophonic responses occurred and contribute to the cortical evoked responses. However, using monopolar stimulations and low rate of stimulation (4Hz) should largely reduce the probability that electrophonic responses contribute to the responses as it has been shown by Sato and colleagues (2016, 2017). This was already discussed in the discussion section on lines 409-420 of the original MS (now lines 473-483).

Please provide more detailed information on how the 441 recordings were derived from 24 guinea pigs and 16-channel electrode arrays. Were measures repeated at different cortical positions in some animals? Please provide a more detailed overview in the method section.
For each animal (n=24), there were between one and five positions (mean 2 positions) of the 16-channel electrode array. For each of these positions we can use between 2-15 recordings in the data analyses (average 9 recordings/position). On average, about 18 recordings per animals were obtained (range: 5 to 41). This is explained at the beginning of the Results section line 287-291.

We have checked that the facilitatory effects of the ramped pulse compared to the rectangular pulses were observed in all animals (lines 416-418 and line 425).

Were electrode sites excluded from the study? How many?
All the cortical recordings that did not provide significant evoked responses to electrical stimulations were excluded. A significant response was an evoked firing rate at least 3 SD above spontaneous activity (lines 252-265 of the revised MS, line 215-217 of the original MS). In total, 768 recordings were collected but only 441 have passed our criteria for a significance response. 

Another issue is, that it is not clear, why all recordings were pooled across animals, and the analysis was not instead performed on the group level of 24 guinea pigs. This is not fully clear. Would the outcome have been the same? Please comment and include an example to support the claim. 
As stated above we have checked that the facilitatory effects observed with the ramped pulses were observed on every animal (lines 416-418 and line 425).

Or at least it should be pointed out more clearly at each instance, how many of the provided neuron numbers (n=x), were stemming from the same/different animal(s) and how many neurons stemmed from the same 16-channel array. Please comment, discuss, and include that information in more detail throughout the results.
For each animal (n=24), there were between one and five positions (mean 2 positions) of the 16-channel electrode array. For each of these positions we can use between 2-15 recordings in the data analyses (average 9 recordings/position). On average, about 18 recordings per animals were obtained (range: 5 to 41). This is explained at the beginning of the Results section line 287-291.

In addition, the number of recordings and of animals involved in each comparison between rectangular pulses and ramped pulses are now provided on the top columns of Table 1 and Table 2.  

What exactly was the trigger level for MUA extraction? Was it an absolute threshold? Please explain.
When adjusting the trigger level individually for each recording site, we aimed at obtaining responses to pure tones from 3-6 Action Potential waveforms. Usually the noise level was below 50µV and the signal-to noise-ratio of the recording was at least 2/1 (action potential amplitude of at least 100µV).
This was added in the Methods section (line 187-189).

Round 2

Reviewer 2 Report

Comments and Suggestions for Authors

The authors adequately addressed all my comments and concerns.